# Association of [^18^F]-FDG PET/CT-Derived Radiomic Features with Clinical Outcomes and Genomic Profiles in Patients with Chronic Lymphocytic Leukemia

**DOI:** 10.3390/diagnostics15101281

**Published:** 2025-05-19

**Authors:** Fabiana Esposito, Luigi Manco, Guglielmo Manenti, Livio Pupo, Andrea Nunzi, Roberta Laureana, Luca Guarnera, Massimiliano Marinoni, Elisa Buzzatti, Paola Elda Gigliotti, Andrea Micillo, Giovanni Scribano, Adriano Venditti, Massimiliano Postorino, Maria Ilaria Del Principe

**Affiliations:** 1Hematology, Department of Biomedicine and Prevention, University of Roma Tor Vergata, 00133 Rome, Italy; andrea.nunzi@ptvonline.it (A.N.); lucaguarnera@live.com (L.G.); massimiliano.marinoni@ptvonline.it (M.M.); buzzattielisa@gmail.com (E.B.); adriano.venditti@uniroma2.it (A.V.); massimiliano.postorino@ptvonline.it (M.P.); dlpmlr00@uniroma2.it (M.I.D.P.); 2Medical Physics Unit, University Hospital of Ferrara, 44124 Ferrara, Italy; luigi.manco@ausl.fe.it; 3Department of Diagnostic Imaging and Interventional Radiology, University of Rome Tor Vergata, 00133 Rome, Italy; guglielmo.manenti@ptvonline.it (G.M.); paolaelda.gigliotti@students.uniroma2.eu (P.E.G.); andrea.micillo@students.uniroma2.eu (A.M.); 4Nuclear Medicine Unit, Department of Oncohaematology, Fondazione Policlinico Tor Vergata, 00133 Rome, Italy; 5Fondazione Policlinico Tor Vergata, 00133 Rome, Italy; livio.pupo@ptvonline.it; 6Postgraduate School in Medical Physics, Physics Department, University of Bologna, 40126 Bologna, Italy; giovanni.scribano2@studio.unibo.it

**Keywords:** chronic lymphocytic leukemia, radiomics, PET/CT, machine learning, genetic profile

## Abstract

**Background:** The role of PET/CT imaging in chronic lymphoproliferative syndromes (CLL) is debated. This study examines the potential of PET/CT radiomics in predicting outcomes and genetic profiles in CLL patients. **Methods:** A retrospective analysis was conducted on 50 CLL patients treated at Policlinico Tor Vergata, Rome, and screened, at diagnosis, with [^18^F]-FDG PET/CT. Potentially pathological lymph nodes were semi-automatically segmented. Genetic mutations in *TP53*, *NOTCH1*, and *IGVH* were assessed. Eight hundred and sixty-five radiomic features were extracted, with the cohort split into training (70%) and validation (30%) sets. Four machine learning models, each with Random Forest, Stochastic Gradient Descent, and Support Vector Machine learners, were trained. **Results:** Progression occurred in 10 patients. The selected radiomic features from CT and PET datasets were correlated with four models of progression and mutations (*TP53*, *NOTCH1*, *IGVH*). The Random Forest models outperformed others in predicting progression (AUC = 0.94/0.88, CA = 0.87/0.75, TP = 80.00%/87.50%, TN = 72.70%/87.50%) and the occurrence of *TP53* (AUC = 0.94/0.96, CA = 0.87/0.80, TP = 87.50%/90.21%, TN = 85.70%/90.90%), and *NOTCH1* (AUC = 0.94/0.85, CA = 0.87/0.67, TP = 80.00%/88.90%, TN = 80.00%/83.30%)mutations. The IGVH models showed poorer performance. **Conclusions:** ML models based on PET/CT radiomic features effectively predict outcomes and genetic profiles in CLL patients.

## 1. Introduction

Chronic lymphocytic leukemia (CLL) is a common hematologic cancer in Western countries, characterized by the proliferation of mature CD5+ B lymphocytes [1].

It is one of the most common forms of leukemia in Western countries, accounting for approximately 25% of new cases every year, with a median age at diagnosis of around 70 years [2,3].

The clinical course of CLL is highly heterogeneous. Most patients have an indolent course, with no need for immediate treatment and prolonged survival, while others present with an aggressive disease requiring early intervention and characterized by frequent relapses [4,5].

Recent advances in understanding CLL pathogenesis have identified crucial prognostic and predictive markers. These discoveries have enabled more precise patient stratification and the development of targeted therapies, significantly altering the traditional treatment approaches [6,7,8,9,10,11].

The evolution of CLL is driven by a complex interplay of genetic factors. Somatic mutations in genes like *TP53*, *NOTCH1*, and *IGHV* are frequently observed and significantly influence disease progression and outcomes [12,13,14,15]. Mutations in *TP53* and unmutated *IGHV* are associated with poor prognosis and disease aggressiveness [16,17]. Other genetic alterations, such as the mutation in the *NOTCH1* genes, has been associated with CLL and may influence prognosis and response to treatment [15,18,19,20].

These genetic changes promote B-cell proliferation and resistance to cell death, leading to the accumulation of neoplastic cells in various lymphoid tissues.

The role of [^18^F]fluorodeoxyglucose positron emission computed tomography ([^18^F]-FDG PET/CT) in chronic lymphoproliferative syndromes and tumor diseases with low uptake is controversial and, to date, no clear consensus has been found on its role in clinical practice [21,22].

Among the various metabolic parameters derived from PET imaging, the most frequently utilized is the maximum standardized uptake value (SUVmax) [23,24].

Several studies, mostly retrospective, have shown that SUV is low in CLL patients, but may increase in patients whose disease transforms into aggressive lymphomas, such as Richter’s syndrome (RS) [21,22,25,26,27,28,29].

Consequently, the role of PET, and its associated SUV parameter, in CLL remains a subject of considerable debate. Its utility has been primarily established in detecting transformation to aggressive lymphomas rather than as a prognostic parameter. For this reason, PET is not routinely employed in the staging of newly diagnosed CLL, with computed tomography (CT) being the standard imaging modality in this context [25,26,27,28,29].

In addition to SUVmax, which reflects the metabolic activity of tumor tissue and does not account for tumor size and volume, the most commonly used volumetric parameters in aggressive lymphoproliferative disease are metabolic tumor volume (MTV) and total lesion glycolysis (TLG). TLG is calculated as the product of the mean SUV and the MTV value [30,31,32,33,34,35]. These semi-quantitative volumetric parameters are recognized as consistent predictors of patient outcomes in various lymphoma subtypes [36,37].

Beyond these parameters, quantitative parameters obtainable through the radiomics process have garnered increasing interest in recent years.

Radiomics is an emerging field, with the potential to play a significant role in cancer research and healthcare studies [38]. Advances in radiomic analysis offer potential implications across the spectrum of clinical practice, including diagnosis, treatment planning, and prognosis, leading to improvement in precision and personalized medicine [32,39,40,41]. The hypothesis behind radiomics is that quantitative image features related to lesion shape, morphology, and heterogeneity reflect the biological and morphologic properties of the tumor. Indeed, the uptake pattern of [^18^F]-FDG recapitulates several biological characteristics of a tumor lesion: vascularization, cellularity, hypoxia, metabolism, cell density, and necrosis [42,43].

Radiomics alone can only give information on well-defined regions of interest. In this context, the use of artificial intelligence and machine learning (ML) has improved the diagnosis of a variety of diseases [44,45]. Radiomic analysis has garnered increasing attention for its capacity to generate predictive models through ML. The main purpose of radiomics is to extract quantitative data (i.e., radiomic features) from multi-modal medical images that are undetectable by the human eye and can be reproduced, interpreted, and correlated with certain clinical endpoints [46].

The use of PET/CT radiomic features (RFts) extracted at baseline in patients with lymphoma has shown promise in improving prognosis by enabling the assessment of tumor heterogeneity, which has been associated with treatment outcome [32,33,34,35,47,48,49]. The possible number of radiomic features extracted from PET and CT images is virtually unlimited, including histogram features, texture features, and features based on patterns, transformations, and shape, which are obtained by various extraction algorithms [50,51,52,53]. While scientific evidence and studies on the usefulness of radiomics in aggressive lymphomas exist in the literature, there are no data or studies addressing this in chronic lymphoproliferative syndromes [32,33,34,35].

In the present analysis, we evaluated, in patients with CLL at disease onset, the correlation between radiomic PET parameters and the genetic profile of the disease and how these might impact the outcome of these patients.

The combination of radiomic features with metabolic and volumetric parameters can ultimately lead to the development of a considerable dataset. The most robust and optimal signatures represent the subset of features most often correlated with clinical outcome.

In this retrospective study, in light of the different targets analyzed—i.e., *TP53*, *NOTCH1*, *IGVH* mutational status and progression—there are different emerging predictive models. For each outcome, two models were built: one with CT-based RFts and the latter with PET-based.

The aim of our study is, therefore, to investigate whether there may be a correlation between radiomic parameters extracted from PET or CT and poor prognosis, mutational profile or disease progression. Thus, the aim is to investigate whether radiomics can provide us with potentially important indications regarding patient outcome.

## 2. Materials and Methods

### 2.1. Population Study

A retrospective analysis was performed on a cohort of CLL patients who were admitted to our institute between January 2016 and April 2022. The minimum follow-up period for patients was 42 months after treatment. In order to make the study population homogeneous, we chose the following inclusion criteria: age greater than 60, patients eligible for therapy with Bruton’s tyrosine kinase inhibitor (BTKi), and [^18^F]-FDG PET/CT imaging.

The database initially included 83 CLL patients aged 60–84 years. However, 33 patients were excluded due to technical imaging issues, radiological examinations not having been performed at our institution, or failure to meet the inclusion criteria.

Fifty patients diagnosed with CLL according to the WHO 2016 classification and selected for BTK inhibitor therapy were enrolled.

All patients underwent a PET/CT scan at the Department of Diagnostic Imaging Radiology, Policlinico Tor Vergata, before starting BTKi therapy. Images were suitable for radiomic analysis.

For each patient, various epidemiological parameters including age and sex were recorded. Additionally, information on the presence of a bulky tumor mass, splenic or extranodal involvement, disease stage (according to Rai and Binet classifications), and clinical and laboratory parameters (complete blood count, LDH, and β2-microglobulin levels) was collected [54].

We also reported the date of diagnosis, date of PET/CT scan, start date of BTK inhibitor therapy, previous treatments, response to therapy, date of progression (if present), and date of last follow-up.

Transformation to Richter’s syndrome did not occur in any patient.

For all patients, the genetic profile and mutation status of several genes, including *TP53*, *NOTCH1*, and the *IGVH* mutation status classified as non-mutated (class 0) or mutated (class 1), were assessed. Progression status was also recorded, with patients classified as no progression (class 0) or progression (class 1).

### 2.2. Image Acquisition and Analysis

All patients underwent PET/CT with [^18^F]-FDG, which was carried out according to the present imaging guidelines [52]. A whole-body PET/CT scan was performed 60 min after the i.v. injection of 3.7 MBq/kg of [^18^F]-FDG with a digital PET/CT scanner (GE Discovery Molecular Insights—DMI PET/CT, GE Healthcare, Waukesha, WI, USA). A free-breathing CT scan was performed from the proximal thigh to the skull base; this was used for attenuation correction purposes, as well as for the anatomic localization of [^18^F]-FDG uptake. The CT scan was performed using automated dose modulation (range 15–100 mA, 120 kV). Immediately after the CT scan, PET images were acquired covering the identical anatomical region. The PET acquisition time was set to 2 min per bed. All PET images were reconstructed using GE VUE Point FX-S algorithm (VPFX-S), a 3D maximum likelihood ordered subset expectation maximization (3D OSEM) image reconstruction algorithm using TOF information and PSF modeling with 4 iterations, 8 subsets and a 6 mm post-processing filter. Datasets were reconstructed with a 256 × 256 pixel matrix.

Two board-certified nuclear medicine physicians reviewed the images using Advantage 4.7 software (GE). Any lymph node exhibiting tracer uptake higher than the background, and not categorized as physiological, was considered potentially pathological. For each patient, the sites and number of lymph nodes with pathological uptake were documented. Baseline PET/CT parameters included the number of FDG-avid lymph nodes, mean and maximum standardized uptake values (SUVmean and SUVmax), total metabolic tumor volume (MTV), and total lesion glycolysis (TLG), with the latter being calculated as MTV × SUVmean. For feature extraction and radiomic analysis, the lymph node with the highest uptake (SUVmax) was designated as the “index lesion” and segmented using a 42% SUVmax threshold employing the aforementioned software (Figure 1).

### 2.3. Radiomic Workflow

The binary classification problem consists in the prediction of different clinical outcomes. Thus, considered targets were defined as follows: progression of CLL, genomic mutation in *TP53*, *NOTCH1* and IGVH. The four targets generated four distinct datasets, with distributions based on 0 (non-progression or unmutated) and 1 (progression or mutated). The four different datasets were analyzed independently, and no normalization or scaling was performed. In this framework, the following pipeline was followed on the four different datasets. Quantitative radiomic features were extracted from each VOI on both PET and CT images separately using the Radiomics package and 3D Slicer image computing platform, according to IBSI standardization [52]. One hundred and twenty-one RFts were extracted from the original images and CT and PET images separately. These RFts are divided in several classes: 14 RFts belong to the original image and mask, 14 to the Shape (3D) class, 18 to First-Order intensity statistics, 24 to Gray-Level Co-occurrence Matrix (GLCM), 16 to Gray-Level Run Length Matrix (GLRLM), 16 to Gray-Level Size Zone (GLSZM), 14 to Gray-Level Dependence Matrix (GLDM) and 5 to Neighboring Gray-Tone Difference Matrix (NGTDM). In addition, 744 textural RFts were extracted from wavelet-decomposed VOIs.

### 2.4. Statistical Analysis and Model Building

The correlation between clinical parameters, [^18^F]-FDG metabolic and volumetric PET/CT parameters was investigated among the different targets group, i.e.,progression, *TP53*, *IGVH*, and *NOTCH1*. The analysis was performed for each VOI using the two-tailed Wilcoxon–Mann–Whitney U-type test. A filter feature selection step, performed using a hand-crafted algorithm in the Python Version 3.12 language, was used to identify robust CT and PET RFts based on the Wilcoxon–Mann–Whitney U-type test. A *p*-value < 0.05 was considered significant in selecting the most robust and non-redundant RFts. This analysis was independently performed for both CT-based and PET-based RFts datasets. Selected robust RFts and clinical parameters were used to build multiple PET and CT ML models. The population was split into training–test (70%) and validation (30%) datasets. The Synthetic Minority Over-Sampling Technique (SMOTE) was used to balance the sample distribution of the considered targets.

Orange Data Mining, an open-source toolkit, was used to train, test, and validate three different learners: RF, SGD, and SVM. All the learners were tested via 10-fold CV using 70% of each of the four datasets. The validation step was conducted on the remaining 30% of the set-aside dataset. Performance models were evaluated in terms of area under the receiving operator characteristic curve (AUC), classification accuracy (CA), precision, sensitivity, specificity, true positive predicted (TP), and true negative predicted (TN). The receiving operating characteristic (ROC) curves were plotted. The performance metrics were calculated in the validation step. The ROC curves were plotted visually.

In the results section, only the models exhibiting the best classification performance for each prediction objective are presented.

## 3. Results

### 3.1. Study Population

The population enrolled in this study included 18 females and 32 males, with an age range from 60 to 84 years. All the 50 patients were staged according to RAI stage, whereby 6 patients were defined as RAI 0, 18 RAI I, 16 RAI II, 6 RAI III and 4 RAI IV.

According to Binet’s classification, 18 patients were A, 20 B and 12 C. The distribution among the different targets and the corresponding descriptive analysis of the clinical, volumetric and metabolic parameters are shown in Table 1. The data collected are reported as mean and standard deviation.

The Wilcoxon–Mann–Whitney U-type test was performed by analyzing the clinical, volumetric and metabolic PET/CT parameters between the different stratified targets. None of these parameters showed statistically significant differences between the different targets, i.e., all *p*-values >0.05. Table 2 shows the *p*-value calculated for each continuous variable between the different groups of clinical metabolic and volumetric parameters.

### 3.2. Radiomic Analysis

The features were selected according to the radiomic algorithm with the aim of finding the most robust and non-redundant subset of RFts highly correlated with the target variable. In this context, four subsets were selected, one for each target, and this procedure was applied to both CT and PET models independently.

From the progression CT and PET datasets, three and seven RFts were selected, respectively; this is the subset with the fewest optimal features. In the *TP53* cases, 16 and 9 RFts were selected in the CT- and PET-based features, respectively. Taking into account *NOTCH1* targets, we obtained two subsets of 10 and 7 RFts in CT and PET, respectively. Lastly, analysis of the *IGVH* dataset yielded two subsets of 16 and 9 RFts in CT- and PET-based signatures, respectively. In every case, the chosen RFts belong to the wavelet-based class. The selected RFts and their corresponding *p*-value are shown in the Appendix A.

### 3.3. Machine Learning Models’ Performances

Comparing three different algorithms, i.e., Random Forest (RF), Stochastic Gradient Descent (SGD), and Support Vector Machine (SVM), the RF classifier was the best in terms of performance in almost all cases. The RF classifier achieved AUC (area under the ROC curve) > 0.80 and CA (classification accuracy) > 0.80. The only exception was in the *IGVH* prediction model, where the accuracy and predictive performance of SGD were better than the corresponding RF in predicting the mutation. The AUC of SGD in this latter case was above 0.75. These performances demonstrate the predictive power of wavelet RFts in gene mutation and progression. Table 3 presents the detailed results for each target across 10-fold CV.

A 30% portion of each dataset was reserved for validation. The ML prediction model demonstrated good performance, with TP and TN percentages well above 85% for the AUC metric. Notably, the *TP53* RF classifier in the PET model achieved the best classification results: AUC = 0.96, CA = 0.80, TP = 90.21%, and TN = 90.9%. Table 4 provides a detailed summary of the validation results.

ROC curves were generated to visualize the performance of the models. For better clarity, the ROC curves were grouped based on the modality (CT or PET) and evaluation method (10-fold CV or validation) (Figure 2).

Finally, a radial plot was built to realize a visual comparison of the performance scores of the CT and PET models (Figure 3).

## 4. Discussion

The role of PET/CT in chronic lymphoproliferative syndromes is controversial and, to date, it has not always been performed in staging and to determine response to treatment [22,25].

To date, there have been few studies evaluating the role of PET in CLL and most focus on the role of PET in detecting possible transformation to Richter’s syndrome [21,22,25,26,27,28,29].

We know that CLL is a low-glucose-uptake lymphoproliferative syndrome in PET and, therefore, in studies in the literature, a correlation between SUV and aggressive tumour or patient outcome has not been found. The role of PET is different when transformation into aggressive lymphoma is suspected and, therefore, in rare cases of transformation into Ricther’s syndrome, PET has an important role, as in all aggressive lymphomas, both in staging and at the end of treatment for response assessment [21,25,26,27,28,29].

We investigated whether there was a correlation between SUV (divided into max, peak and mean SUV and spleen SUV) and mutational profile by focusing on *TP53* mutation, *IGVH* chain hypermutation and *NOTCH* and, from our study, there is no evidence of a statistically significant correlation. This is in line with most studies in the literature and supports the idea that SUV alone is not supportive in CLL but can play a role in the transformation into aggressive lymphoma [21,25,26,27,28,29].

We also did not find a correlation between parameters such as age, presence of bulky mass, LDH and beta2 microglobulin and disease progression or mutational profile.

To ensure population homogeneity, we only enrolled patients starting the same therapeutic regimen. Consequently, the decision to include all eligible BTKi candidates mitigated the potential bias resulting from different therapeutic responses between different therapeutic regimens.

Our analysis of radiomic volumetric variables such as MTV and TGL also showed no statistically significant correlation with outcome in terms of progression and mutational profile, which we know has a worse outcome.

This is in line with the disease’s pathology, which is often clinically diffuse. As it is among the chronic, non-aggressive lymphoproliferative syndromes, the disease is characterised by low glucose uptake, low LDH and widespread lymphadenomegaly [5].

The radiomic and ML prediction analyses in this work underscore the potential role of radiomics in CLL research. The performances obtained in predicting gene mutations are promising, supported by high AUC, accuracy, and other relevant metrics.

The feature selection process, based on a non-parametric filter test, resulted in wavelet-based RFts derived from filtered images. This outcome can be attributed to the unique ability of wavelet transformations to capture spatial and frequency information across multiple scales, enabling the representation of both subtle and prominent variations in the imaging data. By decomposing images into frequency bands, wavelet transformations minimize the influence of noise and artifacts, enhancing the stability of the results.

The ML models were trained using 70% of the dataset, with performance evaluated on a 30% validation set.

Our comparative analysis of three machine learning algorithms—RF, SGD, and SVM—highlighted the superior performance of the RF classifier in most predictive tasks. RF consistently achieved high metrics, with AUC and CA values exceeding 0.80 in nearly all models. The only exception was observed in IGHV mutation prediction, where the SGD classifier outperformed RF, reaching an AUC above 0.75, suggesting its potential suitability for this specific genomic target.

These findings underscore the strong predictive capacity of wavelet-based radiomic RFts, particularly in modeling gene mutations and disease progression. The RF classifier demonstrated high sensitivity and specificity across tasks, as reflected by the TP and TN rates exceeding 85% in most models. Notably, the RF model targeting *TP53* mutations based on PET-derived features yielded the best results, with an AUC of 0.96 and a balanced classification accuracy (CA = 0.80, TP = 90.21%, TN = 90.9%).

This finding is interesting, as we know that patients with *TP53* mutation have a worse prognosis and, therefore, are likely to have intrinsic features that can be detected by radiomics and not by simple qualitative analysis of PET or CT.

The robustness of these models, evaluated through 10-fold cross-validation and confirmed in a 30% hold-out validation set, suggests that ML-based radiomic analysis may provide valuable non-invasive biomarkers for genetic characterization and risk stratification in chronic lymphocytic leukemia.

In our study, both the CT and PET models proved to be highly specific and sensitive in correlating RFts with progression and mutations of *TP53* and *NOTCH1*, showing high accuracy in predicting the mutational profile. One of the most challenging objectives was predicting *IGVH*, where both the PET and CT models exhibited certain limitations in predictive performance.

The accuracy of gene mutation predictions in this study encourages the application of ML tools in clinical practice. As we move forward, the expansion of multicenter studies will be crucial, and further research in CLL is essential. Studies with larger, multicenter populations are needed to better standardize radiomic parameters.

In this study, both the PET and CT models proved effective and there was no marked superiority of either model. Therefore, currently, the use of PET in CLL is not justified. Additionally, it would be valuable to explore not only separate assessments of radiomic parameters in CT and PET but also to develop an integrated PET/CT model.

## 5. Study Limitations

This study presents some limitations that should be acknowledged. Firstly, it is a retrospective, single-center analysis with a relatively small sample size, which may affect the statistical power and limit the generalizability of the findings. Secondly, the absence of an external validation cohort restricts the ability to assess the reproducibility and robustness of the identified radiomic features and their associations with clinical and genomic parameters, thereby currently limiting the applicability of these models in clinical practice. Moreover, image analysis was based on a single target lesion per patient, defined as the most metabolically active lesion (i.e., the one with the highest SUVmax). While this approach improves consistency and feasibility within a hand-crafted radiomics framework, it fails to capture the heterogeneity of lesions that is often characteristic of CLL.

Certainly, this pilot study needs to be confirmed by larger multi-center studies and represents a possible pivotal point for future studies.

## 6. Conclusions

Our findings provide initial evidence correlating PET and/or CT functional parameters with the mutational profile of CLL patients. This first radiomic study of chronic lymphoproliferative syndromes suggests the potential of radiomics in identifying patients with poorer prognosis. Future integration of biological and radiological parameters could improve CLL risk stratification and inform clinical trial design. External validation is crucial to confirm model robustness and generalizability for broad clinical application. This study supports the exploration of intrinsic tumor characteristics predictive of prognosis through standardized and integrated PET and/or CT radiomic analysis, potentially aiding personalized therapy in the future.

## Figures and Tables

**Figure 1 diagnostics-15-01281-f001:**
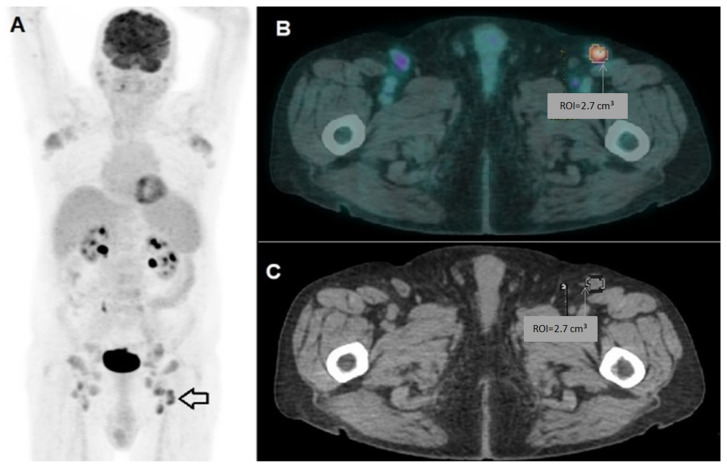
Example patient from the analyzed cohort. (**A**) Whole-body PET/CT showing multiple lymph nodes with pathological radiotracer uptake; the lesion with the highest SUVmax (7.2) in the left inguinal region (arrow) was identified as the index lesion. (**B**) SUVmax-based segmentation of the index lesion on fused axial PET/CT images and (**C**) on CT images.

**Figure 2 diagnostics-15-01281-f002:**
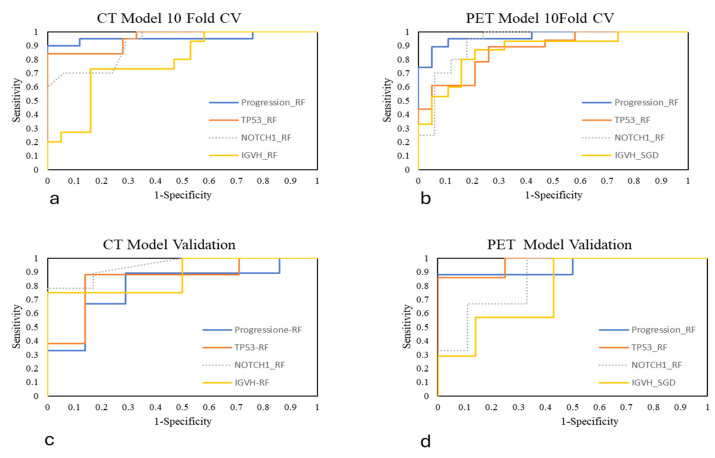
ROC curves in CT model 10-fold CV(**a**), PET model 10-fold CV (**b**), CT model validation (**c**) and PET model validation (**d**). RF: Random Forest; SGD: Stochastic Gradient Descent.

**Figure 3 diagnostics-15-01281-f003:**
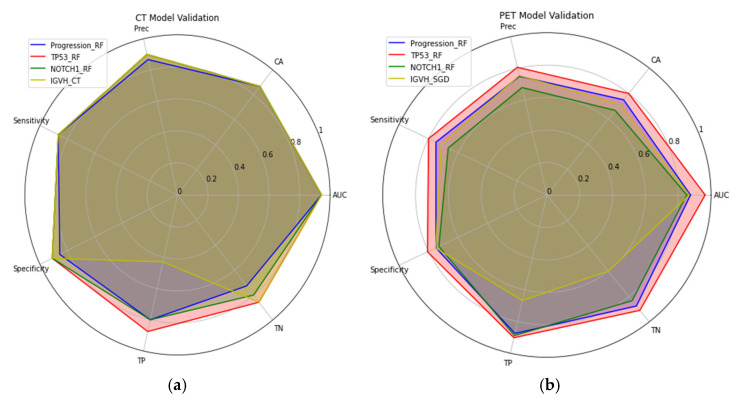
Radial plots of performance scores in CT model validation (**a**) and PET model validation (**b**).

**Table 1 diagnostics-15-01281-t001:** Population characteristics among the different targets.

Patient Characteristics	PR_0	PR_1	TP53_0	TP53_1	NOTCH1_0	NOTCH1_1	IGVH_0	IGVH_1
Number of patients	40	10	38	12	38	12	35	15
Age [year]								
Mean ± St.dev.	63.13 ± 12.94	57.53 ± 8.76	61.36 ± 12.01	63.99 ± 13.85	63.66 ± 10.55	56.52 ± 16.45	60.8 ± 13.57	64.82 ± 8.44
SUV max	3.87 ± 2.6	5.83 ± 3.79	4.58 ± 3.14	3.29 ± 1.91	4.42 ± 3.29	3.78 ± 1.18	4.77 ± 3.19	3.07 ± 1.78
SUV peak	1.66 ± 1.09	2.44 ± 1.59	1.93 ± 1.31	1.45 ± 0.87	1.88 ± 1.38	1.62 ± 0.44	2.04 ± 1.31	1.3 ± 0.84
SUV mean	2.34 ± 1.52	3.51 ± 2.28	2.73 ± 1.84	2.11 ± 1.27	2.66 ± 1.95	2.34 ± 0.62	2.87 ± 1.85	1.89 ± 1.22
MTV	33.8 ± 96.42	29.35 ± 59.46	35.42 ± 97.99	24.64 ± 56.41	35.36 ± 101.38	24.84 ± 27.75	42.43 ± 105.19	9.96 ± 11
TLG	79.89 ± 235.23	53.6 ± 69.34	87.64 ± 238.38	31.69 ± 67.2	78.71 ± 238.19	60.73 ± 85.32	99.14 ± 248.37	15.29 ± 18
Spleen SUV max	2.93 ± 0.81	2.93 ± 0.95	3.01 ± 0.7	2.68 ± 1.15	2.84 ± 0.85	3.24 ± 0.69	3.01 ± 0.76	2.73 ± 0.97
Spleen max diam. [cm]	14.85 ± 3.46	13.43 ± 1.27	14.85 ± 3.4	13.63 ± 2.2	14.87 ± 3.36	13.56 ± 2.35	13.79 ± 2.08	16.4 ± 4.53
Lymph nodes/bulky max diam. [mm]	41.48 ± 28.68	36.57 ± 13.53	43 ± 28.61	32.25 ± 14.37	37.35 ± 20.58	50.63 ± 39.55	45.13 ± 28.91	29.3 ± 13.43
β2-microglobulin levels	3.03 ± 1.85	3.32 ± 1.04	3.03 ± 1.7	3.29 ± 1.85	3.18 ± 1.59	2.8 ± 2.14	3.34 ± 1.8	2.49 ± 1.36
LDH [U/L]	231.25 ± 95.17	333 ± 195.61	254.18 ± 138.8	245.75 ± 76.1	250.46 ± 125.37	257.84 ± 135.93	277.2 ± 140.88	192.2 ± 41.01

LDH: lactate dehydrogenase; MTV: metabolic tumor volume; PR_0: no progression; PR_1: progression; *TP53*_0: unmutated; *TP53*_1: mutated; NOTCH1_0: unmutated; NOTCH1_1: mutated; *IGVH*_0: unmutated; *IGVH*_1: mutated; SUV: standardized uptake value; TLG: total lesion glycolysis.

**Table 2 diagnostics-15-01281-t002:** *p*-Value results for Wilcoxon–Mann–Whitney U-type test in clinical, metabolic and volumetric parameters.

Parameter	PR	TP53	NOTCH1	IGVH
Age [year]	0.33	0.58	0.25	0.58
SUV max	0.33	0.24	0.64	0.24
SUV peak	0.42	0.24	0.50	0.24
SUV mean	0.34	0.28	0.57	0.28
MTV	0.78	0.12	0.08	0.12
TLG	0.46	0.10	0.20	0.10
Spleen SUV max	0.97	0.36	0.26	0.36
Spleen max diam. [cm]	0.58	0.45	0.28	0.45
Lymph nodes/bulky max diam. [mm]	0.86	0.60	0.40	0.60
β2-microglobulin levels	0.39	0.68	0.43	0.68
LDH	0.12	0.79	0.34	0.79

LDH: lactate dehydrogenase; MTV: metabolic tumor volume; PR: progression; SUV: standardized uptake value; TLG: total lesion glycolysis.

**Table 3 diagnostics-15-01281-t003:** ML performances scores in 10-fold CV.

CT_10-Fold CV	AUC	CA	Precision	Sensitivity	Specificity
Progression_RF	0.96	0.92	0.92	0.92	0.93
TP53_RF	0.95	0.81	0.81	0.81	0.81
NOTCH1_RF	0.91	0.81	0.84	0.81	0.83
IGVH_RF	0.78	0.68	0.69	0.68	0.69
**PET_10-Fold CV**					
Progression_RF	0.96	0.90	0.90	0.90	0.90
TP53_RF	0.87	0.81	0.82	0.81	0.82
NOTCH1_RF	0.92	0.87	0.87	0.87	0.86
IGVH_SGD	0.84	0.77	0.79	0.77	0.79

AUC: area under the curve; ML: machine learning; RF: Random Forest, SGD: Stochastic Gradient Descent.

**Table 4 diagnostics-15-01281-t004:** ML performance scores in validation.

CT Model	AUC	CA	Precision	Sensitivity	Specificity	TP	TN
Progression_RF	0.94	0.87	0.87	0.87	0.86	0.80	0.73
TP53_RF	0.94	0.87	0.90	0.87	0.91	0.87	0.86
NOTCH1_RF	0.94	0.87	0.90	0.87	0.91	0.80	0.8
IGVH_RF	0.94	0.87	0.90	0.87	0.91	0.43	0.86
**PET Model**							
Progression_RF	0.88	0.75	0.75	0.75	0.75	0.87	87.50
TP53_RF	0.96	0.80	0.81	0.80	0.81	0.90	0.91
NOTCH1_RF	0.85	0.67	0.68	0.67	0.73	0.89	0.83
IGVH_SGD	0.87	0.71	0.76	0.71	0.75	0.67	0.60

Machine learning (ML); area under the curve (AUC); classification accuracy (CA); true-positive (TP); true-negative (TN); Random forest (RF); Stochastic Gradient Descent (SGD).

## Data Availability

The data presented in this study are available on request from the corresponding authors.

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
