# Peer review of "Association of [18F]-FDG PET/CT-Derived Radiomic Features with Clinical Outcomes and Genomic Profiles in Patients with Chronic Lymphocytic Leukemia"

_diagnostics, 2025, doi:10.3390/diagnostics15101281_

Round 1
Reviewer 1 Report
Comments and Suggestions for Authors
Please consider section order. Standard may be Introduction, Methods, Results, Discussion.
The current Introduction section is overly long and should be shortened to improve readability and focus.
Line 308: Replace "scan" with "scanner."
Methods Section: Add a subsection describing Statistical Analysis.
The method for calculating SUVmin (mentioned in Table 2) is missing.
Line 328: Please clarify the basis for the reported 42%.
Nomenclature Consistency: Both [18F]FDG and [18F]-FDG exist.
Table 1: Indicate the unit for LDH.
Table 2: "AGE" may be "Age (year)".
Table 4: Only TP and TN values shown as percentages.
Author Response
Please, see the attachment.

Reviewer 2 Report
Comments and Suggestions for Authors
The manuscript by Esposito and colleagues explores the association between PET-based radiomic features and clinical outcomes in patients with Chronic Lymphocytic Leukemia (CLL).
Specifically, the study aims to compare four machine learning models and evaluate the most effective tool for predicting disease progression and the presence of mutations.
Overall, the article presents both strengths and weaknesses. It is generally well-written, with the methods and results sections being well described and easy to follow. However, it's a radiomics study conducted on only 50 patients (training set - 15 patients) in a clinical setting where 18F-FDG PET imaging is not particularly useful. Additionally, sections of the manuscript appeared in some point disconnected.
I suggest making the following revisions before considering the manuscript for possible publication:
Introduction:
- Somewhat scattered and at times redundant—please streamline. It feels as though some parts are disconnected, with logical jumps. Please rewrite the introduction in order to follow a clear, logical sequence. Summarize certain concepts and eliminate unnecessary phrases.
- Clearly state the aim of the study
Methods:
The section " population study" appeared disconnected to the rest of the manuscript: please revised.
- A 42-month cutoff was used as part of the selection criteria—what is the rationale behind this choice? Please add a reference if available.
- The text mentions patients selected before initiating BTK inhibitors, yet these are not discussed elsewhere in the manuscript. Please clarify.
- Table 1 (patients characteristics) is not reported and referred to another table.
- Regarding the segmentation phase: it seems that features were extracted from an index lesion. Segmenting only one lesion is limiting in light of the heterogeneity of the disease and does not capture its full complexity. If no alternative is feasible, please explicitly state that only the “index lesion” was considered for both semiquantitative parameters and radiomic features.
Discussion:
- Avoid unnecessary sentences, e.g., lines 222–225 and 247–250.
- Add references for lines 236–238 (mentioned twice, likely to emphasize importance).
- Discuss the results and model comparisons in greater detail.
- Include a paragraph addressing the limitations of the study.
- Make the conclusion more concise and to the point.
Minor Points:
- Check all abbreviations and superscripted mass numbers throughout the text.
Author Response
Please, see the attachment

Round 2
Reviewer 2 Report
Comments and Suggestions for Authors
The authors consistently revised the manuscript according to the suggestions. The paper is now suitable for publication in the Diagnostics journal.